# Successional dynamics of a 35 year old freshwater mitigation wetland in southeastern New Hampshire

J. Grant McKown[1]*, Gregg E. Moore[1], Andrew R. Payne[2], Natalie A. White[3], Jennifer L. Gibson[1]

1 Department of Biological Sciences and Jackson Estuarine Laboratory, School of Marine Science and Ocean Engineering, University of New Hampshire, Durham, New Hampshire, United States of America, 2 Department of Biodiversity, Earth, & Environmental Sciences, Academy of Natural Sciences of Drexel University, Philadelphia, Pennsylvania, United States of America, 3 Department of Natural Resources, University of New Hampshire, Durham, New Hampshire, United States of America

* jgrantmck@gmail.com

**Data Availability Statement:** All relevant data are contained within the manuscript and its Supporting information files.

## Abstract

The long-term ecological success of compensatory freshwater wetland projects has come into question based on follow-up monitoring studies over the past few decades. Given that wetland restoration may require many years to decades to converge to desired outcomes, long-term monitoring of successional patterns may increase our ability to fully evaluate success of wetland mitigation projects or guide adaptive management when needed. In Portsmouth, New Hampshire a 4 ha wetland was constructed in an abandoned gravel quarry as off-site compensatory mitigation for impacts to a scrub-shrub swamp associated with property expansion. Building upon prior evaluations from 1992 and 2002, we conducted a floral survey in 2020 to compare results with prior surveys to document vegetation successional trends over time. In addition, we monitored the avian community throughout the growing season as a measure of habitat quality. The plant community mirrored documented successional trends of freshwater wetland restoration projects as native hydrophytes dominated species composition. Plant species composition stabilized as the rate of turnover, the measurement of succession, declined by nearly half after 17 years. Researchers should consider long-term monitoring of specific sites to better understand successional patterns of created wetlands as we documented long time frames required for the development of scrub-shrub swamps, red maple swamps, and sedge meadows. High species richness was attributed to beaver activity, topographic heterogeneity from *Carex stricta* tussocks, and the seed bank from the application of peat from the original wetland. Habitat heterogeneity of open water, herbaceous cover, and woody vegetation supports a diverse avian community including 11 wetland dependent species. Although the mitigation project has not created the full area of lost scrub-shrub swamp after 35 years, it has developed a structurally complex habitat and diverse avian community that effectively provides the functions and values of the impacted system.

**Funding:** As stated in the acknowledgements, JGM received funding from the University of New Hampshire Graduate School Student Teaching Assistant Fellowship in support of his Master's Thesis Research. The funders had no role in study design, data collection and analysis, decision to publish, or preparation of the manuscript.

**Competing interests:** The authors have declared that no competing interests exist.

## Introduction

The goal of wetland mitigation in the United States under the Section 404b program of the Clean Water Act (1977), the federal government's "no net loss" policy, and New Hampshire's Fill and Dredge in Wetlands Act (1969) is the creation of a self-sustaining wetland ecosystem equal in size, structure, and function to the one which was lost [1]. Ecologists and regulators have sought to improve the likelihood of ecological success of mitigation by identifying potential causes of failure such as lack of government accountability [2, 3] and improper establishment of hydrology [4] as well as improvements to restoration strategies such as seeding [5, 6] and incorporation of microtopography [7, 8]. The Section 404b program innately assumes, through monitoring periods of less than five years and lack of universally required management strategies, that mitigation projects will remain ecologically successful long after meeting regulatory success criteria. However, recent studies have revealed declining ecological success of the freshwater vegetation communities 6–10 years after wetland construction [4, 9], questioning the sustainability of compensatory mitigation as an effective tool compared to avoidance and minimization of wetland impacts.

One of the challenges to restoring ecologically equivalent freshwater mitigation wetlands is the inability to accurately predict the trajectory of the vegetation community. Matthews and Spyreas [10] proposed a framework to interpret how a created wetland may converge to or diverge from restoration goals. Convergence results in species composition and relative abundances eventually resembling reference conditions, though the path may be linear or non-linear. Divergence is the process of reaching an alternative stable state, and may occur initially or after a considerable amount of time following restoration efforts. A common documented trajectory is an initial convergence, as early colonizers and annuals are outcompeted, and then a divergence due to the lack of uncommon perennials or formation of alternative wetland communities [10]. For example, Aronson and Galatowitsch [11], through repeated surveys over twenty years, described prairie pothole systems reaching an alternative stable state to references after initially converging with the accumulation of common emergent and floating aquatics. Within 12 years the wetlands had generally stabilized into alternative stable states with lower species richness, lack of representative wet prairie and woody species, and invasion of aggressive exotics.

It has been proposed that the vegetation community of wetlands mature within 15–20 years [12] based on stabilization of species composition and richness and declines in the rate of species turnover. Proposed by Noon [13] first and amended by follow-up studies [10, 14, 15], species composition of the vegetation community is immediately dominated by annuals and shifts over time to perennials and dominant graminoids like *Typha* and *Sparganium*. For example, Deberry and Perry [16] observed annuals comprised 60% of the species of 2-year-old created wetlands compared to 4% of adjacent reference wetlands. Second, species richness reaches a maximum between 5–15 years and subsequently plateaus or declines [5, 17]. A decline in species richness has been explained by (1) a lack of equal recruitment of perennial species to annual species loss [15, 18], (2) hydrophytes outcompeting upland species after proper hydrology establishment [19], and (3) invasive species outcompeting natives and forming monocultures [11, 20]. Third, the rate of species gain, loss, and turnover, measured as the rate species are lost and replaced over time, is initially high after wetland creation and then declines over time [11, 15, 21]. Although rarely quantified, the rate of species gain, loss, and turnover could describe if a wetland reaches an equilibrium or remains dynamic over time.

Comprehensive, long-term floristic reviews of a site can provide needed context to understand wetland successional dynamics and vegetation restoration trajectories. A common experimental design involves surveying a chronosequence of created wetlands over one or two

growing seasons to draw conclusions [22]. The experimental design is cost-effective and has allowed researchers to understand how wetlands recover with a short turnaround. However, large variability between sites has been documented in multiple studies, where site context (inherent factors of landscape management history, landscape location, climate, etc.) has proven to be just as a significant factor in vegetation dynamics compared to landscape, hydrology, or wetland size parameters [15, 17]. Specific site context could provide more details pertinent to understanding the development of unique wetland communities. Consistent, long-term floristic reviews of individual sites could pinpoint mechanisms that explain shifts in restoration trajectories.

In addition to proper habitat structure, a common goal for wetland creation is the support and enhancement of avian communities for conservation or recreation [23]. Species composition and distribution of the vegetation community are likely important factors in predicting a wetland's avian community. The diversity and abundance of the avian community can be sensitive to certain vegetation and landscape metrics including wetland area size [24], forested landscape cover [25], structural complexity and heterogeneity within the wetland [24, 26], and open water cover [27]. Glisson et al. [28] found through intensive vegetation sampling efforts that secretive marsh birds were highly sensitive to vegetation metrics. For example, *Porzana carolina* (Sora) preferred greater *Typha* cover while the invasion of *Phalaris arundinacea* deterred use by *Botaurus lentiginosus* (American bittern) and *Rallus limicola* (Virginia rail). Short and long-term shifts in avian use have been attributed to changes in a wetland's vegetation community [29, 30]. The avian community then is a function of the successional dynamics of the vegetion community and may serve as metrics of habitat quality and mitigation success.

With the federal government's Section 404b program and New Hampshire's Fill and Dredge in Wetlands Act approaching 50 years of age, researchers now have the ability to assess long-term successional dynamics, mechanisms of succession, and the ecological success of created wetlands. Evaluation of long-term restoration trajectories (e.g., $\geq$ 30 years) could determine whether vegetation communities eventually stabilize and converge to desired conditions or diverge to less desired, alternative stable states [10, 31]. Towards this goal, our study analyzed how the vegetation community of a 35 year old created wetland in southern New Hampshire shifted over time based on three floristic surveys over the last 28 years. The goals of the study were to (1) compare long-term successional trends of the wetland to documented patterns in the literature, (2) determine if the vegetation community converged towards a scrub-shrub swamp or diverged to an alternative state, and (3) evaluate the quality of habitat for wetland birds. We conducted a floristic survey in 2020 of the vegetation community and incorporated species composition and habitat delineation data from 1992 and 2002 to analyze successional patterns. Additionally, we monitored the avian community over one season as an assessment of habitat quality.

## Materials and methods

### Site description

A 4 ha freshwater wetland was created in an abandoned gravel pit mine, Quarry Pond (43.0234, -70.8004), in the winter of 1985–86 in Portsmouth, New Hampshire as off-site compensatory wetland mitigation after the destruction of a similar sized scrub-shrub swamp resulting from site infrastructure expansion [32]. Seven pools were excavated to the groundwater table. Peat from the original wetland was excavated, containing an intact seed bank, and deposited at the restoration site. The peat was spread at a thickness of 15–30 cm across the restoration area. Surface water runoff, direct precipitation, and groundwater are the primary

water inputs into the topographically restricted restored wetland. Several output channels for seasonal flooding connects downgradient to Packer Bog, a 121 ha unfragmented forested swamp northwest of the site [33]. The dominant plant community of Packer Bog is a regionally rare Atlantic white cedar–yellow birch–pepperbush swamp. The revegetation plans relied solely on natural colonization and the seed bank from the excavated peat. Descriptions of the vegetation community from the original wetland and first seven years of the Quarry Pond project are unknown [32, 34]. The species composition was documented in 1992 and 2002 [34, 35] and wetland communities delineated in 1992. Beaver activity was noted for raising the water levels by 1 m after blocking outlet channels [32, 34]. Beaver activity was presumed to have been consistent over time and was also observed in 2020, represented by the presence of three beaver lodges, evidence of foraging on *Alnus* shrubs, and creation of new channels.

## Vegetation sampling and analysis

Vegetation was surveyed for species composition and distribution of distinct wetland plant communities at the site in two ways. To complete a full inventory of species present, we conducted meander surveys throughout the wetland up to 3 hours biweekly from late May to late October. Meander survey efforts were conducted at rougly the same duration of season and frequency as in 1992 and 2002 surveys. Voucher specimens for each species were collected and accessioned in the New Hampshire Archive at the Albion H. Hodgdon Herbarium at the University of New Hampshire. Vegetation nomenclature is based on Haines et al. [36]. In addition to meander surveys, we completed fixed plot linear transect surveys in June. Vegetation was surveyed every 10 m along four linear transects positioned perpendicular to the southeastern boundary. Each transect was 240–290 m in length resulting in a total of 103 plots. Visual cover was estimated to the nearest 1% (maximum of 100%) at three canopy layers: understory (0.5 $m^2$ square quadrat), shrubs of 2–5 m (3 m radius), and trees greater than 5 m tall (5 m radius) (modified from Spencer et al. [37]). The wetland community type was classified for each plot according to Sperduto and Nichols [38] based on ground, shrub, and tree species composition and cover. Vegetation plot sampling efforts were based on the methods of the initial 1992 survey to accurately compare the distribution of vegetation communities, however the exact locations of the transects and plots from 1992 are unknown. The visual cover and distribution of vegetation communities of 2020 were only compared to 1992, since 2002 survey methodology did not allow for direct comparison.

Approximately 120 high resolution images were captured on June 1, 2020 (Fig 1) utilizing a DJI (Los Angeles, CA) Phantom 4 Pro Unmanned Aerial Vehicle (UAV) at an altitude of 200ft and equipped with a DJI 20MP true color (R, G, B) camera. UAV imagery had approximately 80% overlap and at an effective 1.67cm ground resolution. Imagery capture was conducted within two hours of solar noon to maintain consistency of environmental conditions. Resulting images were mosaicked together using Agisoft photogrammetry software and then rectified to ground coordinates. The R, G, B mosaic bands were then stacked and clipped to the bounds of the study area utilizing ArcGIS Pro 2.5 software (ESRI, Redlands, CA). UAV flight planning was completed using DJI Flight Planner software. Flight control was completed with DJI Ultimate Flight v3 and Drone Deploy software (San Francisco, CA).

Using the resulting mosaics and plot sampling data, the areal extents of each wetland community were manually digitized using ArcGIS Pro. The community boundary delineations were groundtruthed by field verification. The dominant vegetation delineations of 1992 were reclassified according to Sperduto and Nichols [38], georeferenced based on permanent locations of center of pools and wetland perimeter, and manually digitized. The areas of the wetland community types in 1992 and 2020 were calculated based on digital delineations in

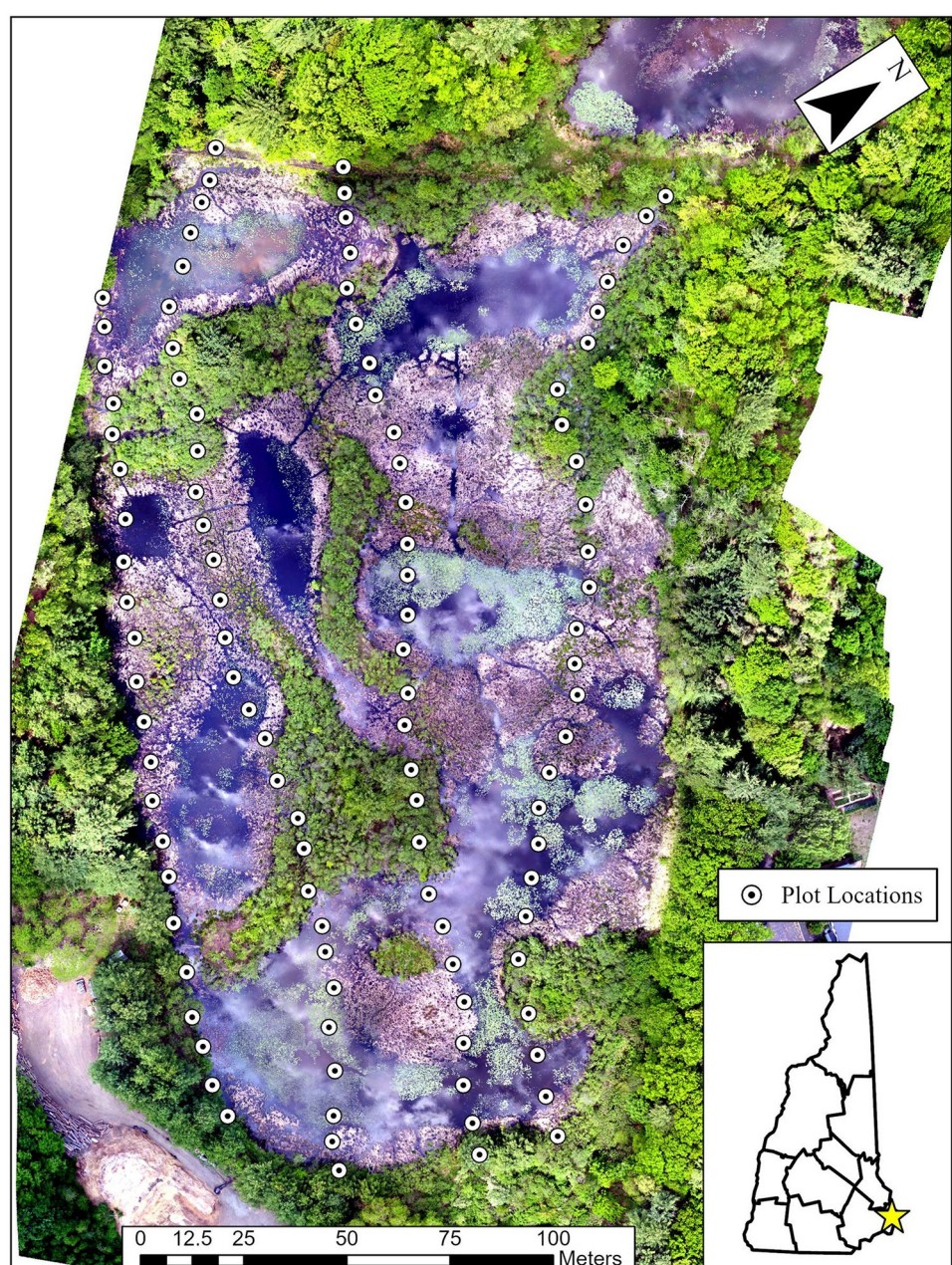

**Fig 1. UAV imagery and vegetation sampling plot locations at Quarry Pond in Portsmouth, New Hampshire.**

ArcGIS Pro. The habitat delineation maps of 1992 and 2020 were compared to assess wider community shifts over the past 28 years. Plot-level understory species richness was compared between different wetland communities by aggregating the eight communities based on similar hydrologic conditions and dominant vegetation: aquatic bed, cattail marsh (cattail, cattail–scrub-shrub marsh), graminoid meadow marsh (tall graminoid, tall graminoid–scrub-shrub, and sedge meadow marsh), and woody swamps (scrub-shrub, red maple swamp). Species richness data were power transformed to meet the assumption of normality per the Shapiro-Wilk Test. One-way ANOVA and follow-up Tukey-Kramer tests were used to compare plot-level

species richness between the aggregated groups in JMP 15 software (SAS Institute, Inc., Cary, NC).

To determine how plant communities have changed over time, the species composition surveys of 2020 were compared to 1992 and 2002 [34, 35]. Species were assigned wetland indicator designations, native status, growth habits, life history, and coefficients of conservatism. The United States Department of Agriculture (USDA) PLANTS database [39] was used to classify native status, growth habit (forb/herb, graminoid, and woody), and life history (annual and perennial) for all species. The wetland indicator score (WIS), the probability of the species occurring in a wetland environment, was assigned according to the Northeast Region of the National Wetland Plant List of 2016 [40]. Each WIS is assigned a rank value: obligate (OBL) = 1, facultative wet (FACW) = 2, facultative (FAC) = 3, facultative upland (FACU) = 4, and upland (UPL) = 5. A non-weighted Prevelance Index (PI) was calculated based on WISs to quantitatively assess shifts in hydrophyte and upland species. The lower index values correspond to a presence of more hydrophytic vegetation, with values 1–2 representing a dominance of OBL vegetation. The PI was calculated as:

$$PI = \frac{\sum WIS}{S}$$

where S is the number of species in each sampling year.

The coefficient of conservatism (CoC), a measure of a species's tolerance to anthropogenic disturbance, was based on the New Hampshire assessment by Bried et al. [41]. CoCs are assigned collaboratively by by regional expert botanists and values range from zero to ten, where zero is highly tolerant to disturbance or an exotic and ten is a species intolerant of distrubance [42]. A floristic quality assessment (FQA) was conducted for each survey to assess the change in conservation value over time. The FQA is a quantitative masurement of a site's relative lack of human disturbance [43] and has often been used as a factor in success criteria for mitigation and habitat assessments [44, 45]. The FQA is calculated as:

$$FQA = C_{site} * \sqrt{S}$$

where $C_{site}$ is the mean CoC of an entire site and S is the species richness of the site.

Sorensen's Index of Similarity (Sorensen Index) assessed the difference in community composition between the floristic surveys to understand dynamic changes in the vegetation community [46]. The Sorensen Index compares the presence/absence of species between two samples. Sorensen's Index is calculated as:

$$Sorensen's\ Index = \left(\frac{2c}{a+b}\right) * 100$$

where a is the number of species found in the first sample, b is the number of species found in the second sample, and c is the number of species shared between the two samples.

The annual rate of species gain, loss, and turnover from 1992 to 2020 was calculated according to Anderson [27]. Turnover, measured as the rate species are lost and replaced over time, provides insight into whether species composition is stabilizing. [15]. The annual proportional

species gain ($G_p$), loss ($L_p$), and turnover ($T_p$) was calculated with presence/absence data as:

$$G_p = \frac{G}{(0.5)(S_{t_1} + S_{t_2})(Dt)} (yr^{-1})$$

$$L_p = \frac{L}{(0.5)\left(S_{t_1} + S_{t_2}\right)(Dt)} (yr^{-1})$$

$$T_p = \frac{G + L}{(0.5)(S_{t_1} + S_{t_2})(Dt)} (yr^{-1})$$

where $S_t$ is species richness, G is the number of species observed in $t_2$ not observed in $t_2$, and L is the number of species in $t_1$ not observed in $t_2$, and $\Delta t$ is the time interval between surveys. Based on our intensive sampling efforts, it was assumed that a species not found in a survey had disappeared from the wetland and was not an artifact of sampling. Additionally, $G_p$ was calculated for the 1985–1992 period based on the assumption that no wetland species were present immediately after restoration due to lack of initial plantings [32, 34].

## Avian community sampling and analysis

The avian community was surveyed from early June to late October 2020, encompassing the late breeding, nesting, and fall migration seasons of wetland-associated and wetland-dependent birds [47]. COVID-19 related restrictions on research activities at the University of New Hampshire prevented surveying during recommended breeding and spring migration seasons of northern New England [48]. The distribution of *Alnus*, *Frangula*, and *Rhus* shrubs on the constructed islands created natural barriers dividing the wetland into distinct zones. Surveyors created four permanent vantage points along the perimeter and an additional point on an immediately adjacent, hydrologically-connected pond where vegetation was not surveyed. The size of the birding zones ranged from 0.29–0.87 ha, and the furthest visual extent within each zone ranged from 75–120 m.

We conducted repeated point count surveys to estimate species richness and relative abundance [48]. Bird surveys were conducted biweekly (at least 10 days apart) and completed between 0700 and 0845 hours. Point count surveys were conducted in the same order with the same personnel every visit. Avian surveys were conducted outside of the wetland, at a distance sufficient to not alter the behavior of the birds. Surveys were not conducted during fog, rain, heavy rain or with loud noise from an adjacent lumber yard. Repeated point count surveys consisted of a 10 minute passive visual and audial surveys [47]. All individuals seen or heard breeding, nesting, foraging, or resting in the wetland, including the upland islands, or on the immediate edge were recorded.

Each bird species was classified as wetland-dependent, wetland-associated, or upland based on the classification of Brooks and Croonquist [49]. Bird species assigned a score of 5 were classified as wetland-dependent, 3 as wetland-associated, and 1 or 0 as upland. The species richness, wetland richness, mean abundance, diversity, and evenness were calculated for the avian community [48]. Diversity was calculatedas effective diversity (D), which is based on the Shannon-Weiner Diversity Index (H'):

$$D = e^{-\sum p_i \ln(p_i)}$$

where $p_i$ is the relative abundance of each bird species across all of the summer and fall surveys

[50]. Eveness of the community was calculated using Pielou's J ($J'$):

$$J' = \frac{H'}{H'_{max}}$$

where $H'$ is the Shannon-Weiner Diversity Index and $H'_{max}$ is the idealized $H'$ of the community where all species are equally abundant [51].

## Results

### Plant species composition shifts

A total of 129 plant species, comprising 54 families and 85 genera, were identified in the 2020 floral survey (S1 Table). The community experienced an increase of 19 species, 14 genera, and 6 families since the last survey in 2002 (Table 1). Despite a net increase in richness, 21 species observed in 2002 were not observed in the 2020 survey. Additionally, 9 of those species had been present in both 1992 and 2002. The survey identified 12 species that were found in the 1992 yet absent in 2002. Consistent observations have been made across the three survey intervals for two New Hampshire state listed endangered species, *Cardamine bulbosa* and *Potamogeton foliosus*, and one threatened species, *Sparganium eurycarpum* [52]. In our most recent survey, *Typha* x *glauca* was not observed but it's likely that this cryptic hybrid remains due to

**Table 1. Descriptions of the vegetation community from each floristic survey of Quarry Pond.**

| Vegetation Metric | 1992 | 2002 | 2020 |
|---|---|---|---|
| **Community Description** | | | |
| Family | 46 | 48 | 54 |
| Genera | 67 | 71 | 85 |
| Species | 101 | 110 | 129 |
| **Conservation Value** | | | |
| Average Coefficient of Conservatism | 3.65 ± 0.15 | 3.95 ± 0.16 | 3.63 ± 0.16 |
| Floristic Quality Assessment | 36.7 | 41.5 | 41.2 |
| **Wetland Status** | | | |
| Prevalence Index | 1.41 | 1.38 | 1.62 |
| Wetland | 101 | 110 | 121 |
| Upland | 0 | 0 | 8 |
| **Native Status** | | | |
| Native | 96 | 97 | 93 |
| Exotic | 4 | 3 | 7 |
| **Life History** | | | |
| Annual | 14 | 9 | 14 |
| Perennial | 86 | 91 | 86 |
| **Growth Habit** | | | |
| Forb/herb | 60 | 60 | 60 |
| Graminoid | 28 | 25 | 23 |
| Woody | 12 | 15 | 17 |

Wetland species consist of those classified as OBL, FACW, and FAC. Upland species consist of those classified as FACU and UPL. Mean ± standard error reported for coefficient of conservatism.

The percent of the community is reported for each classification of wetland indicator status, native status, life history, and growth habit.

**Table 2. Descriptions of the species gained between floral surveys of Quarry Pond.**

| Metric | 1992–2002 | 2002–2020 | 1992–2020 |
|---|---|---|---|
| New Species | 31 | 40 | 47 |
| **Conservation Value** | | | |
| Average Coefficient of Conservatism | 4.61 ± 0.33 | 3.38 ± 0.34 | 4.37 ± 0.27 |
| **Wetland Status** | | | |
| Prevalence Index | 1.32 | 2.15 | 1.98 |
| Wetland | 31 | 32 | 39 |
| Upland | 0 | 8 | 8 |
| **Native Status** | | | |
| Native | 30 | 34 | 43 |
| Exotic | 1 | 6 | 4 |
| **Life History** | | | |
| Annual | 3 | 8 | 8 |
| Perennial | 28 | 32 | 39 |
| **Growth Habit** | | | |
| Forb/herb | 18 | 21 | 28 |
| Graminoid | 8 | 10 | 7 |
| Woody | 5 | 9 | 12 |

Wetland species consist of those classified as OBL, FACW, and FAC. Upland species consist of those classified as FACU and UPL. Mean ± standard error reported for Coefficient of Conservatism.

The number of species are reported for each classification of wetland indicator status, native status, life history, and growth habit.

dense neighboring stands of *T. angustifolia* and *T. latifolia* and its documented presence in 2002.

The species composition of the wetland remained generally similar over the observation period from 1992 to 2020 despite species turnover at each interval. Net changes at the 2002 and 2020 observation intervals were seemingly minor, despite measurable losses and subsequent gains (Tables 2 and 3), *i.e.*, there was a net gain of 9 new species from 1992–2002; then another net gain of 19 species from 2002–2020. The inclusion of the 2002 survey illustrates steady trends of turnover.as the Sorenson Index remained steady between 71–75% over the past 28 years (Table 4).

The wetland plant community remained primarily composed of natives, perennials, hydrophytes, and forbs throughout the 28 years of monitoring. Perennials continually dominated after seven years post-construction, fluctuating between 86–91% of the community. Natives remained above 90% for all surveys. Forbs interestingly represented exactly 60% of the community for all three surveys. The proportion of graminoids steadily declined by 5% as woody species replaced them. Upland species had a marked increase from 0% to 6% of the community from 2002 to 2020. The eight upland species were represented by a lone individual thus not contributing significantly to the plant community. These species were exclusively found at the upper edges of the wetland on top of *Carex stricta* tussocks. The inclusion of these species in the 2020 survey contributed to the shift in PI from 1.38 to 1.61 between 1992 to 2020.

The conservation value of the wetland complex peaked in 2002 and then declined slightly by 2020 (Table 1). The average CoC of the community peaked at 3.95 in 2002 before declining to 3.63, the lowest value of all three surveys. The species gained and lost between 2002 and 2020 had average CoCs of 3.38 and 4.86, respectively. The FQA peaked at 41.5 in 2002 and only declined slightly by 2020 (Table 1). The FQA was supported by an increase in species

**Table 3. Descriptions of the species lost between floral surveys of Quarry Pond.**

| Metric | 1992–2002 | 2002–2020 | 1992–2020 |
|---|---|---|---|
| Lost Species | 22 | 21 | 19 |
| **Conservation Value** | | | |
| Average Coefficient of Conservatism | 3.50 ± 0.38 | 4.85 ± 0.34 | 3.87 ± 0.31 |
| **Wetland Status** | | | |
| Prevalence Index | 1.41 | 1.38 | 1.37 |
| Wetland (OBL + FACW) | 22 | 21 | 19 |
| Upland (FAC + FACU + UPL) | 0 | 0 | 0 |
| **Native Status** | | | |
| Native | 19 | 21 | 19 |
| Exotic | 3 | 0 | 0 |
| **Life History** | | | |
| Annual | 7 | 0 | 4 |
| Perennial | 15 | 21 | 15 |
| **Growth Habit** | | | |
| Forb/herb | 13 | 10 | 12 |
| Graminoid | 9 | 7 | 5 |
| Woody | 0 | 4 | 2 |

Wetland species consist of those classified as OBL, FACW, and FAC. Upland species consist of those classified as FACU and UPL. Mean ± standard error reported for Coefficient of Conservatism.

The number of species are reported for each classification of wetland indicator status, native status, life history, and growth habit.

richness despite a decline in the average CoC. The losses of highly senstivie species and gains of exotic and generalists drove the declines in conservation values.

The rates of change for the wetland community declined over time driven by decreases in species losses and turnover (Table 4). $G_p$ (annual rate of species gain) was calculated as 0.286 yr$^{-1}$ for the initial seven years post-construction. $G_p$ decreased dramatically after the initial seven years to 0.029 yr$^{-1}$ in 1992 to 2002 and then continued to decline in 2002 to 2020. $L_p$ (annual rate of species loss) decreased by over half from 0.021 yr$^{-1}$ to 0.010 yr$^{-1}$ in 2002 to 2020. $T_p$ (annual rate of successional turnover), followed the same pattern decreasing from

**Table 4. Rates of species gains, loss, and turnover of the vegetation community of Quarry Pond.**

| Metric | 1986–1992 | 1992–2002 | 2002–2020 | 1992–2020 |
|---|---|---|---|---|
| **Species Shifts** | | | | |
| Similar Species | | 79 | 89 | 82 |
| New Species | 101 | 31 | 40 | 47 |
| Lost Species | | 22 | 21 | 19 |
| **Annual Proportional Rate of Change** | | | | |
| Gp—Species Gain (yr$^{-1}$) | 0.333 | 0.029 | 0.019 | 0.015 |
| Lp—Species Lost (yr$^{-1}$) | | 0.021 | 0.010 | 0.006 |
| $T_p$—Species Turnover (yr$^{-1}$) | | 0.050 | 0.028 | 0.020 |
| **Community Similarity** | | | | |
| Sorenson Similarity (%) | | 74.9 | 74.5 | 71.3 |

Sorenson Index is a measure of the similarity of the vegetation community between two surveys.

0.050 yr $^{-1}$ to 0.028 yr $^{-1}$, respectively. Overall, the average $T_p$ for the wetland complex was 0.020 yr $^{-1}$ for 1992 to 2020, suggesting low but steady rates of successional turnover after seven years post-construction.

## Wetland plant community shifts

The floral survey of 2020 detailed a structurally complex and heterogeneous temperate freshwater wetland. We delineated eight wetland communities described by Sperduto and Nichols [38]: cattail marsh, aquatic bed, scrub-shrub swamp, mixed tall graminoid–scrub-shrub marsh, tall graminoid meadow marsh, seasonally flooded red maple swamp, emergent marsh, and sedge meadow marsh. Several areas consisted of a transition stage between cattail marsh and scrub-shrub swamp and were described as mixed cattail–scrub-shrub marsh as a ninth habitat. The upland islands remained a prominent feature and were dominated by *Rhus typhina* shrubs and several mature *Pinus strobus* trees. There was a significant difference in plot-level understory species richness across the four aggregated community groups ($F_{3, 91}$ = 11.9, p < 0.001). Species richness was divided into two tiers of high richness in sedge meadow marshes (5.7 ± 0.6 SE) and woody swamps (5.1 ± 0.5) and lower richness cattail marshes (3.1 ± 0.3) and aquatic bed pools (2.7 ± 0.2).

The wetland communities were dynamic with major changes in areal extent and distribution in the past 28 years (Figs 2 and 3). The wetland complex has become more heterogeneous as the number of communities increased from four (cattail marsh, tall graminoid meadow marsh, marshy moat, and aquatic bed) to nine. The only community to have not persisted was the marshy moat subcommunity in the southwest portion of the site. The cattail marsh expanded its range by replacing 42% and 37% of the tall-graminoid meadow marsh and aquatic bed areas of 1992, respectively. The total area of cattail marsh increased by 80% as the aquatic bed and tall graminoid meadow marsh decreased by 38% and 80%, respectively (Table 5). The areal coverage of the wetland complex expanded by 0.09 ha due to the development of the sedge meadow marsh and red maple swamp outside the original construction boundaries.

There was a shift of dominant species within the cattail marsh and aquatic bed communities since 1992. The understory herbs and forbs of cattail marshes had shifted from *Juncus effusus*, *Lemna minor*, and *Phalaris arundinacea* to *Nuphar variegata* and *Carex* sp. (Table 6). The expansion of cattail into permanent shallow waters was demonstrated by the presence of *N. variegata* intermixed within cattail reeds. In the aquatic bed pools, floating leaf aquatics shifted from *Potamogeton natans*, *Potamogeton pusillus*, and *Chara* sp. to a community dominated by *Brasenia schreberi* and *N. variegata*. The 2002 survey noted declines in *Chara* sp. and rise of *B. schreberi*, and the pattern has continued in the last 18 years. The dominant bladderwort species, *Utricularia gibba*, was also replaced with *Utricularia macrorhiza*, which was widespread and the only bladderwort species found in 2020.

The graminoid meadow marsh communities experienced steep declines in area, shifts in geographical distribution, and turnover of dominant vegetation. In 1992, the meadow marshes would have been classified as tall graminoid meadow marshes due to the dominance of *Carex stricta* or co-dominance of *Phalaris arundinacea* and *Juncus effusus*. Cattail marsh had replaced *Phalaris-Juncus* meadow marshes by 2020, continuing a pattern noted in 2002. Additionally, an emergent marsh has replaced the *C. stricta* marsh in the south. The tall graminoid meadow marsh community continued, however, in the form of scattered *C. stricta* patches. The mixed tall graminoid–scrub-shrub form of the community dominate at the upland-wetland ecotone along the outer perimeter and shores of interior uplands, where light gaps in the shrub canopy supports a *C. stricta* understory. The mixed tall graminoid–scrub-shrub marshes

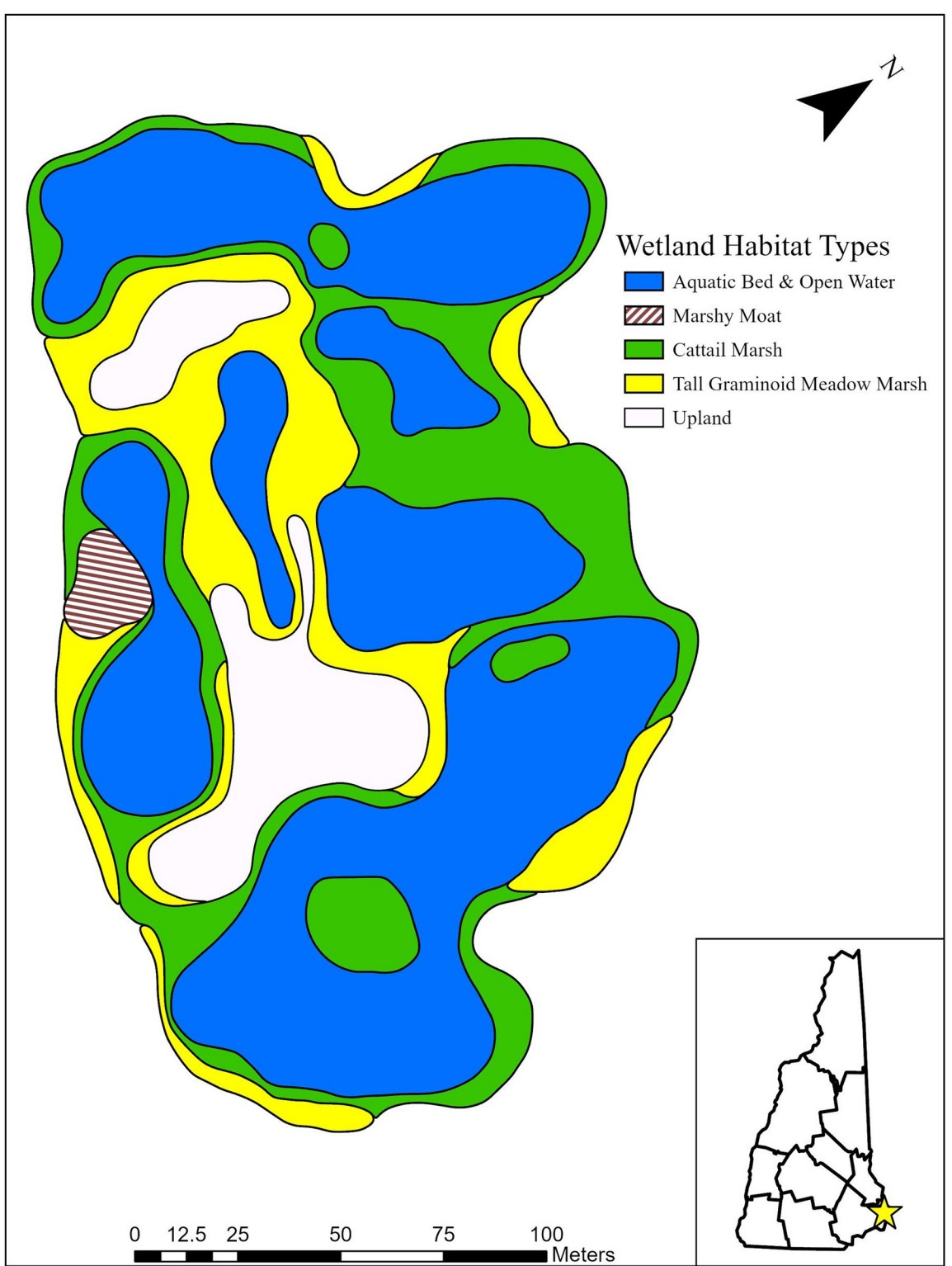

**Fig 2. Wetland community distribution map of Quarry Pond in 1992 based on map and descriptions from Padgett and Crow 1994.**
Wetland community types were reclassified from dominant vegetation classification, georeferenced to drone imagery, and manually digitized in ArcGIS Pro. Wetland channels were not described in detail in 1992.

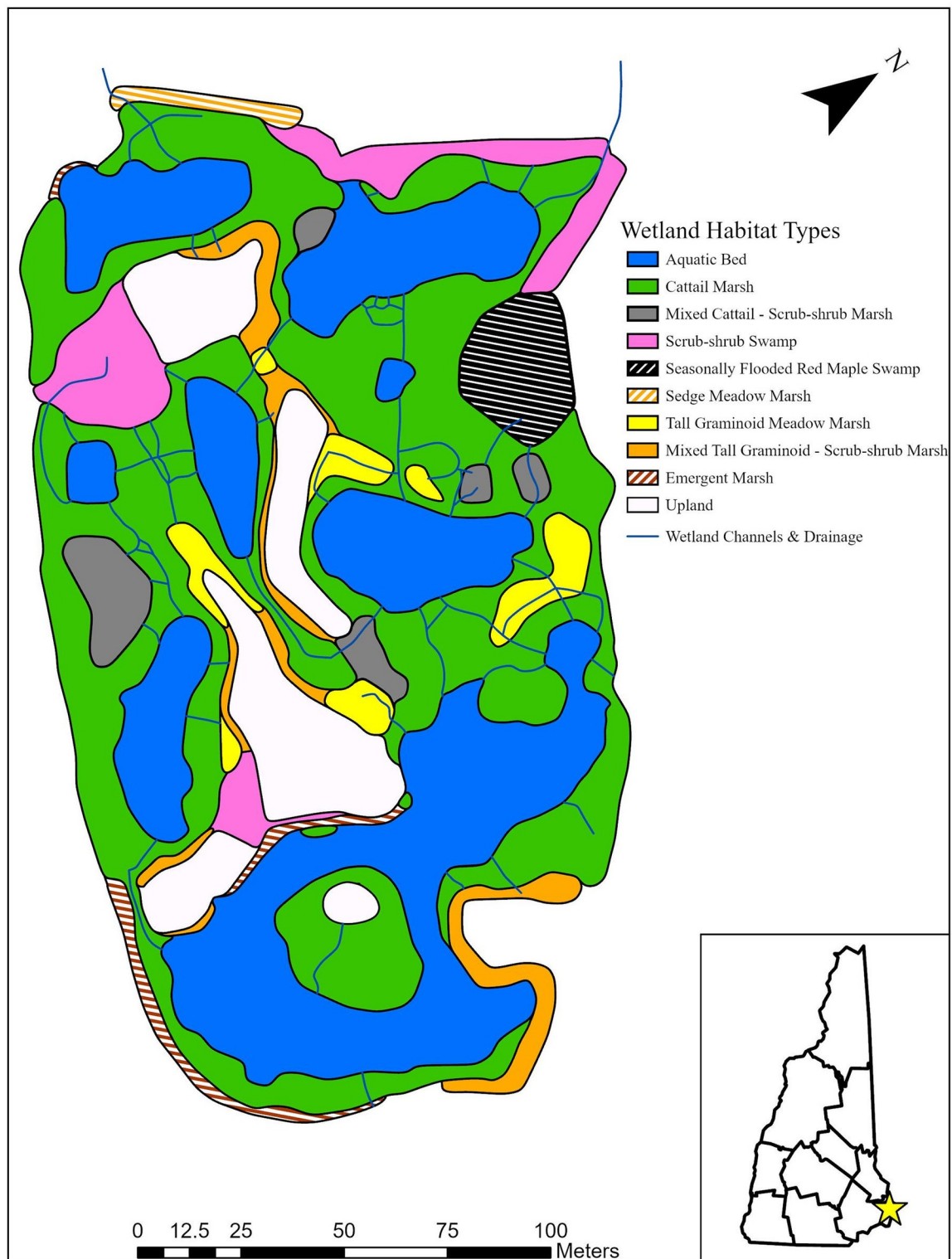

**Fig 3. Wetland community distribution map of Quarry Pond in 2020 based on plot sampling and drone imagery.** Wetland community types were classified based on plot sampling of ground cover, shrub, and tree canopy. Areas and wetland channels were manually digitized using drone imagery in ArcGIS Pro.

**Table 5. Change in areas (ha) of wetland communities of Quarry Pond from 1992 to 2020.**

| Community Type | 1992 Area (ha) | 2020 Area (ha) |
|---|---|---|
| Aquatic Bed & Open Water | 1.45 | 0.91 |
| Cattail Marsh | 0.64 | 1.15 |
| Mixed Cattail—Scrub-shrub Marsh | 0.00 | 0.08 |
| Scrub-shrub Swamp | 0.00 | 0.18 |
| Seasonally Flooded Red Maple Swamp | 0.00 | 0.08 |
| Sedge Meadow Marsh | 0.00 | 0.02 |
| Tall Graminoid Meadow Marsh | 0.46 | 0.09 |
| Mixed Tall Graminoid—Scrub-shrub Marsh | 0.00 | 0.13 |
| Emergent Marsh | 0.00 | 0.04 |
| Marshy Moat | 0.04 | 0.00 |
| Interior Upland | 0.29 | 0.29 |
| **Total Wetland Area (ha)** | **2.59** | **2.67** |

Areas were calculated in ArcGIS Pro based on the digitized wetland communities. Interior upland area was excluded from total wetland area calculation.

were not recorded in the first two floral surveys. The combination of both forms of tall gramminoid meadow marshes still represent only 47% of the original area.

The increase in shrub and tree canopy cover led to the development of four new wetland community types and a widspread presence of woody vegetation at the site over time. The combination of all communities with a shrub and tree canopy element would represent the third largest wetland community of 0.47 ha, a major increase since 1992. Previous floral studies did not mention any notable extent of shrub and tree cover. *Alnus incana* and *Frangula alnus* shrubs have expanded into the wetland, especially into cattail marsh demonstrated by the presence of the mixed cattail–scrub-shrub marshes. The narrow mixed tall graminoid—scrub-shrub marshes with *C. stricta* understory has replaced prior interior uplands or *Phalaris-Juncus* meadow marshes. The southeastern patch of scrub-shrub swamp has formed on a prior upland island, potentially driven by the creation of a inlet channel by beavers. Lastly, the red maple

**Table 6. The average mean cover and plot frequency of understory species in the cattail marsh and aquatic bed communities of the 2020 floristic survey of Quarry Pond.**

| Species | Visual Cover | Plot Frequency (%) |
|---|---|---|
| **Cattail Marsh** | | |
| *Typha latifolia* | 15.3 ± 1.2 | 97 |
| *Carex* sp. | 1.6 ± 0.8 | 16 |
| *Nuphar variegata* | 1.2 ± 0.6 | 13 |
| **Aquatic Bed** | | |
| *Nuphar variegata* | 16.5 ± 4.2 | 59 |
| *Brasenia schreberi* | 15.4 ± 4.4 | 56 |
| *Utricularia macrorhiza* | 5.5 ± 1.7 | 59 |
| *Potamogeton* sp. | 2.0 ± 0.79 | 44 |
| *Potamogeton natans* | 1.4 ± 0.5 | 30 |

Visual cover is reported as mean ± standard error.

All species with an average mean cover greater than 1.0 are shown. *Carex* sp. includes all *Carex* species except *Carex stricta*. *Potamogeton* sp. includes all *Potamogeton* species except *Potamogeton natans*.

swamp formed behind a berm (most likely a relict from construction), replacing previous uplands.

## Exotic species

Two common exotic species, *Phalaris arundinacea* and *Frangula alnus*, followed noteworthy trends over the past 28 years. In 1992, *P. arundinacea* was widespread as *Phalaris-Juncus* meadow marshes comprised 18% of the wetland and the grass occupied 20% of the cover within the community. Although still present in 2020, *P. arundinacea* was sparse being found in only 27% of all non-aquatic plots with a low mean cover of 0.7 ± 0.2 (Table 7). In the shrub layer, *Frangula alnus* was present throughout the wetland, typically coexisting with *Alnus incana*. The exotic shrub exhibited a dramatic expansion since first identified in 2002. *F. alnus* had the highest plot frequency of 54% and second highest mean cover of 7.8 ± 1.3. Additionally, *F. alnus* seedlings and saplings were found extensively in the understory with a plot frequency of 20% and mean cover of 1.0 ± 0.3, posing a future risk of exotic dominance for the shrub canopy. Despite the rise in *F. alnus* abundance, the wetland woody richness only declined by one species from 2002 to 2020.

## Avian survey

The 2020 point count surveys detected 430 individuals comprising 11 wetland-dependent, 3 wetland-associated, and 25 upland species. Monitoring efforts identified *Porzana carolina* and *Rallus limicola* as secretive marsh species using the cattail marsh. *P. carolina* is listed as a species of Special Concern for the State of New Hampshire [53]. Although wetland birds comprised 45% of total individuals, *Agelaius phoeniceus* (Red-winged blackbird) consisted of 68% of all wetland individuals when flocks of over 30 birds were observed throughout June and July

**Table 7. Mean cover and plot frequency of common species of understory and shrub canopy across all non-open water plots of the 2020 floristic survey of Quarry Pond.**

| Species | Visual Cover | Plot Frequency (%) |
|---|---|---|
| **Understory** | | |
| *Typha latifolia* | 9.7 ± 1.0 | 72 |
| *Carex stricta* | 8.1 ± 2.2 | 24 |
| *Carex* sp. | 1.4 ± 0.5 | 21 |
| ***Frangula alnus*** | **1.0 ± 0.3** | **20** |
| *Persicaria amphibia* | 0.9 ± 0.3 | 17 |
| *Onoclea sensibilis* | 0.8 ± 0.5 | 13 |
| ***Phalaris arundinacea*** | **0.7 ± 0.2** | **27** |
| *Boehmeria cylindrica* | 0.6 ± 0.3 | 6 |
| *Nuphar variegata* | 0.6 ± 0.3 | 7 |
| ***Lythrum salicaria*** | **0.5 ± 0.1** | **25** |
| **Shrub Canopy** | | |
| *Alnus incana* | 9.7 ± 1.8 | 44 |
| ***Frangula alnus*** | **7.8 ± 1.3** | **54** |
| *Vaccinium corymbosum* | 1.1 ± 0.6 | 6 |
| *Rhus typhina* | 0.9 ± 0.5 | 7 |

Bolded species are classified as exotic in New Hampshire. Visual cover is reported as mean ± standard error.

All species with an average mean cover greater than 0.50 are shown. *Carex* sp. includes all *Carex* species except *C. stricta*.

**Table 8. The five most common wetland (dependent and associated) and upland species of the avian community monitoring in 2020.**

| Common Name | Surveys Present | Total Individuals | Proportion of Community (%) | Mean Abundance (indiv. per survey) |
|---|---|---|---|---|
| **Wetland Species** | | | | |
| Red-winged Blackbird | 6 | 132 | 30.6 | 14.7 ± 7.3 |
| Mallard | 3 | 19 | 4.4 | 2.1 ± 1.3 |
| Wood Duck | 3 | 15 | 3.5 | 1.7 ± 1.0 |
| Common Yellowthroat | 3 | 4 | 0.9 | 0.4 ± 0.2 |
| Great Blue Heron | 4 | 4 | 0.9 | 0.4 ± 0.2 |
| **Upland Species** | | | | |
| Common Grackle | 5 | 41 | 9.5 | 4.6 ± 2.6 |
| Gray Catbird | 9 | 33 | 7.7 | 3.7 ± 0.6 |
| Blue Jay | 9 | 32 | 7.4 | 3.6 ± 0.6 |
| Black-capped Chickadee | 5 | 23 | 5.3 | 2.6 ± 1.0 |
| Song Sparrow | 8 | 21 | 4.9 | 2.3 ± 0.7 |

Mean abundance is reported as mean ± standard error.

(Table 8). The wetland was heavily utilized by the avian community with 47.8 ± 9.3 individuals and 12.1 ± 0.9 species observed per survey. The community recorded effective diversity of 14.81 D and evenness of 0.38 J across the entire season of monitoring.

## Discussion

### Drivers of species composition shifts

Plant species successional trends mirrored prior documented patterns in the literature of life history traits, growth habits, and rates of gain, loss, and turnover. Species gain rates during the first seven years of wetland development resembled early successional patterns for restored wetlands based on high rates of turnover and rapid increases in species richness [5, 15, 54, 55]. Annuals initially dominate the species composition of wetlands and decrease over time from competition from perennials and clonal graminoids [10, 16]. In this study, annuals comprised roughly 10–15% of the community after seven years. It is unknown if annuals dominated the site during the first six years as we do not have published records of site monitoring that might have occurred.

The permanent shifts from annuals to perennial and woody species are in agreement with other long-term wetland successional studies and models [11, 13]. The proportion of different species guilds remained relatively constant as well, except for a small increase in woody species and decrease in forbs. The species composition of the vegetation community is stabilizing and followed documented successional trends based on the species gain, loss, and turnover rates declining by half between 17 and 35 years post-restoration [15, 21]. Despite major shifts in the distribution of wetland communities, over 60% of the original species were found 28 years after the initial floristic survey. Atkinson et al. [14] found that the species composition remained static for 20 year old restored wetlands, attributing the stability to the resilience of dominant perennials.

The seed bank within the peat from the original wetland most likely allowed for rapid colonization and for seeds to persist until suitable germination conditions developed [56]. Wetland seed banks in general are vertically and laterally heterogeneous and reflect long-term hydrological conditions and successional history [57]. Seed banks of prairie pothole systems have been shown to be viable up to 20 years post-draining [58]. Prior restorations utilizing transplanted wetland soils have shown considerable success with high wetland richness and cover

in short time frames [59, 60]. Although not directly studied, the application of the original seed bank in the restoration process probably led to both rapid colonization in the first seven years and long-term stabilization of the species composition.

Topographical complexity within the wetland might have allowed for persistence of specialized flora. Original construction details were aimed at creating pools connected by channels and gentle sloped mounds throughout the wetland. Efforts to create heterogenous hydrological conditions through manipulation of topography, even on centimeter scales, have been shown to increase species richness in restored wetlands [7, 61]. *Carex stricta* tussocks of the meadow marshes can function as natural microtopographic hummocks that provide refugia from prolonged flooding, microhabitat variation in moisture and redox conditions, and greater light penetration [62, 63]. The *C. stricta* tussock meadow marshes had the greatest species richness and functioned as plant diversity hotspots, where certain species like *Hypericum boreale*, *Campanula aparanoides*, and *Scutellaria* sp. exclusively resided.

Species richness in restored wetlands is predicted to peak and decline, or plateau, within 15 years [18, 64]. Species richness in this study continued to increase over time. Upland species partially drove richness gains as their share of the community increased by 6% since 2002. Only one individual was found for all upland species and were all primarily located in the narrow borders of *C. stricta* at the upland-wetland edge. If gains in these upland species are excluded, the species richness in this study seems to be stabilizing as the richness increased roughly the same amount but in double the time (1992–2002 vs. 2002–2020). Woody species likely took advantage of suitable dry soil conditions in the wetland-upland edge from a prolonged drought period spanning 2016–2017 to establish [65]. Upland annuals found in saturated soils in May and June at the wetland edge are more difficult to explain, but likely will not persist with the continuance of typical hydrologic conditions [18].

## Mechanisms of trajectory shifts

From a community perspective, the wetland complex remained dynamic after the proposed maturation period of 15–20 years [12]. The restoration trajectory of the wetland community structure shifted in the past 18 years based on the floral survey of 2002. The shifts could be seen through (1) cattail marsh replacement of aquatic bed and meadow marsh habitat, (2) successional shifts of dominant vegetation within certain communities, (3) development of woody vegetation, (4) formation of smaller and more specialized niches, and (5) divergent trajectories of the invasives *Phalaris aundinacea* and *Frangula alnus*. Possible mechanisms of long-term shifts in restoration trajectory may be alterations of the hydrological regime and shoreline herbivory from beaver activity and nonlinear development rates of different wetland communities.

Beaver activity can induce major vegetation community shifts within riparian zones and existing wetlands by raising water table elevation, increasing shoreline complexity, removing woody vegetation on shorelines, and altering nutrient cycling dynamics [66–69]. Increased flooding, grazing on macrophytes, and tree felling from beaver activity has been attributed to successful wetland restoration projects [70]. Beaver reintroductions on degraded wetlands can increase plant species richness, within site heterogeneity, and community evenness [70, 71]. The impacts of beaver activity may not be fully realized until at least 10 years after colonizing a site [71]. It is possible that beaver activity might have shifted the long-term trajectory of the vegetation community at Quarry Pond by altering the hydrological regime and maintaining a sparse shrub canopy. An increase of seasonal flooding depth created proper hydrologic conditions for the red maple swamp and sedge meadow marsh to develop. Additionally, the consumption of *A. incana* maintains scrub-shrub–tall graminoid meadow marsh community on

shoreline banks by increasing light availability for *C. stricta* and associated herbaceous understory [70]. Rentsch et al. [72] attributed tree felling in scrub-shrub swamps to the persistence of rare understory flora in open light gaps in West Virginia wetlands.

Exotic species invasion is a common factor in assigning failure for restored wetlands to reach regulatory success and ecological parity with natural references. Common invasives of temperate wetland were present in both the understory and shrub layers [73, 74]. Notably, *Phalaris arundinacea* abundance was reduced consierably since 1992 as tall graminoid meadow marsh habitat declined. Although *P. arundinacea* rapidly colonized the wetland complex within seven years, it has not formed dense monocultures, common in wetland restoration projects [75]. The decline of *P. arundinacea* is encouraging for the future of this restoration project since *P. arundinacea* has been shown to reduce species richness [76], prevent the development of graminoid meadow guilds [59, 77], and increase biotic homogenization across landscapes [11, 78]. The natural *P. arundinacea* decline warrants future research given documented negative impacts on wetland restoration projects and expensive, time-consuming control options [79].

The shrub layer of the wetland complex experienced a widespread expansion of *Frangula alnus* since its presence was first documented in 2002. Mills et al. [74] documented a similar expansion within a 20 year timespan in Wisconsin. Possible mechanisms for *F. alnus* invasion are release from disease and herbivory pressure [80], avian dispersal of attractive fleshy fruits [81], and high tolerance for varying soil and hydrologic conditions [82, 83]. Bird species exclusively observed in the exotic shrubs such as *Dumetalla carolinensis* (Gray catbird) and *Cyanocitta cristata* (Blue jay) may be vectors for seed distribution [81]. Additionally, this study confirms previous findings that *F. alnus* develops a dense understory of seedlings allowing for high recruitment of propagules [83]. The wetland woody species remained stable as *F. alnus* canopy cover increased in the past 18 years, suggesting the exotic shrub might not be negatively impacting the vegetation community. However, a lag time remains to be seen between the development of *F. alnus* and decline in shrub species richness [74].

## Avian community as a indicator of habitat quality

The presence of scrub-shrub swamp, cattail marsh, and aquatic bed habitat supported a diverse avian community including 11 wetland-dependent, 3 wetland-associated, and 2 secretive marsh species. Habitat heterogeneity allowed for waterfowl and marsh birds to utilize preferred habitats for foraging, breeding, and roosting. The dominant expanse of cattail marsh supported secretive marsh species *P. carolina* and *R. limicola* and large flocks of *A. phoeniceus* [28, 29]. Wetland species including *Ardea herodias* (Great blue heron), *Butorides striatus* (Green heron), *Aix sponsa* (Wood duck), and *Anas platyrhynchos* (Mallard) were predominantly found in open water and emergent marsh habitat. The formation and maintenance of pools by beavers in wetlands has been documented to increase waterfowl abundance and richness [84, 85]. By raising water levels and grazing, beaver activity might have indirectly benefitted certain waterfowl in this study by preventing further expansion of cattails into pools.

The wetland complex possibly supported the avian community of the immediate upland forests and greater Packer Bog. The most common upland bird species, with the exception of *Quiscalus quiscula* (Common grackle), were found primarily on *Alnus* and *Frangula* shrubs on the wetland edge. *F. alnus* fruits are commonly foraged by the common upland species including *Mimus polyglottos* (Northern mockingbird), *Dumetella carolinensis*, and *Zomotrichia albicollis* (White-throated sparrow) [81]. Additionally, certain upland bird species were predominantly found in wetland habitat such as *Melospiza melodia* (Song sparrow) in cattail

marsh. Hapner et al. [26] documented increases in upland species richness and density in created wetlands over time, attributing the increases to expansion of emergent and woody vegetation. As the vegetation structure shifted the avian community likely responded, especially with increases of those reliant on cattail and woody vegetation [26, 29]. Long-term monitoring of both habitat structure and avian community response would provide a better framework for approaching wetland restoration projects aimed at boosting waterfowl populations.

### Long-term restoration success

Non-linear development of certain wetland communities may initially appear diverging from restoration goals or reference conditions. At Quarry Pond, woody vegetation (scrub-shrub and red maple swamps) and sedge meadow marsh required at least 20 years to develop after increased flooding into the areas from beaver activity. Long-term studies of prairie pothole systems have noted lag times for similar woody vegetation and wet prairie wetland communities [19, 86]. Additionally, Sueltenfuss and Cooper [78] documented that proper hydrology does not lead to similar wetland communities within a watershed even after 15 years. Targeted restoration activities such as planating propagules, seeding, or long-term maintenance may be needed to jump-start, or at least hasten, these late-developing communities and prevent permanent divergence from reference conditions [20, 87].

Quarry Pond has developed a diverse vegetation community with prominent open water, herbaceous, shrub, and tree elements within 35 years since creation. The wetland remained primarily composed of hydrophytic natives. The ecological features of the nine wetland subcommunities now support 121 wetland plant species. *Frangula alnus* is the only prominent invasive species in the wetland. The structurally complex wetland supported a diverse avian community including 14 wetland species that actively utilize different herbaceous, shrub, and tree elements. Despite the development of approximately 0.18 ha of scrub-shrub habitat within the restoration site footprint, this mitigation project has not fully replaced the original 4 ha of intact scrub-shrub swamp lost in 1985 However, if the goal was to create a functioning freshwater wetland which supports diverse, native flora and fauna communities, the Quarry Pond compensatory wetland mitigation project would appear to be successful and self-sustaining given present site conditions. Long-term monitoring and flexible restoration goals to account for possible restoration trajectory shifts should be incorporated in future freshwater mitigation projects to avoid anthropogenic bias that limits assigning success.

## Conclusions

Freshwater wetland mitigation projects should strive to incorporate long-term monitoring and set flexible restoration goals take into account possible shifts in the vegetation community. Restoration practicioners can utilize a destroyed wetland's seed bank for rapid establishment of hydrophytic vegetation while focusing on the construction of mound and pool topography, inclusion of beavers, and enhancement of *C. stricta* presence to increase niche space and species richness. Goals to restore late-developing communities like scrub-shrub swamps or sedge meadows might require more than 20 years to achieve without targeted initial actions and continued maintenance. Long-term, repeated monitoring of sites can provide guidance for future projects by documenting site-specific successional trends and driving mechanisms. It is possible the restoration trajectory of Quarry Pond will converge to the lost scrub-shrub swamp, but nevertheless, the wetland contributes ecological benefits through its functions and values documented herein.

## Supporting information

**S1 Table. Compiled list of plant species presence and absence across three floristic surveys (1992, 2002, and 2020) of Quarry Pond.**
(PDF)

**S2 Table. Description of the entire avian community based on biweekly point count surveys at Quarry Pond in 2020.** Wetland dependency ratings are based on Brooks and Croonquist 1990.
(PDF)

## Acknowledgments

The authors thank Cornerstone Tree Care for providing access the site. Field assistance was provided by graduate student Chloe Brownlie (UNH) for vegetation monitoring. Technical assistance UAV and remote sensing was provided by Taylor Goddard and William Winslow (UNH). David Burdick (UNH) and Tom Ballestero (UNH) provided invaluable feedback on the project proposal and manuscript. Jackson Estuarine Laboratory contribution number 578.

## Author Contributions

**Conceptualization:** J. Grant McKown, Gregg E. Moore.

**Data curation:** J. Grant McKown.

**Formal analysis:** J. Grant McKown, Gregg E. Moore, Andrew R. Payne, Jennifer L. Gibson.

**Funding acquisition:** J. Grant McKown.

**Investigation:** J. Grant McKown, Gregg E. Moore, Andrew R. Payne, Natalie A. White.

**Methodology:** J. Grant McKown, Gregg E. Moore.

**Project administration:** Gregg E. Moore.

**Supervision:** Gregg E. Moore.

**Visualization:** J. Grant McKown.

**Writing – original draft:** J. Grant McKown, Gregg E. Moore.

**Writing – review & editing:** J. Grant McKown, Gregg E. Moore, Andrew R. Payne, Natalie A. White, Jennifer L. Gibson.

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
