## [Decision Letter · Decision Letter 0]

24 Mar 2021

PONE-D-21-04357

Successional dynamics of a 35-year old freshwater mitigation wetland in southeastern New Hampshire

PLOS ONE

Dear Mr. McKown,

Thank you for submitting your manuscript to PLOS ONE. After careful consideration, we feel that it has merit but does not *fully* meet PLOS ONE’s publication criteria as it currently stands. The reviewer (yes, one) and I agree that the manuscript needs revisions, but that they are relatively minor. Congratulations! Many manuscripts (my own included) do not get to that level after several tries. Therefore, we invite you to submit a revised version of the manuscript that addresses the points raised during the review process.

Please see our comments below, and submit your revised manuscript by May 08 2021 11:59PM. Please note that the deadline is system-defined - If you will need more time than this to complete your revisions, please reply to this message or contact the journal office at plosone@plos.org. Please include the following items when submitting your revised manuscript:

We look forward to receiving your revised manuscript.

Kind regards,

David G. Jenkins, PhD

Academic Editor

PLOS ONE

Editor Comments:

Dear Mr. McKown,

As you may know, it is difficult to obtain multiple reviewers for submitted manuscripts these days - beyond the exponential expansion of journals and manuscripts compared to slight increases in authors, the pandemic is restricting reviewer availability. So I was able to obtain one review of your manuscript, and in the interest of moving forward, I dove in as well.

The Good News is that the reviewer considers your manuscript as needing only minor revisions, and I concur. Well done!

In those review comments, I especially point to their comments regarding lines 152-3, 476-8, 569-71. I fully agree with those comments. For example, the long time interval in your study is great, but also bears the need to match methods as carefully as you can for fair comparisons through time. Please explain more fully how you did that, and to what extent methods may have changed, including how transects were located each time (e.g., randomly? matching placement in 2020?). If you matched identically your methods with earlier methods, then saying so will help greatly.

About diagnosis of scrub-swamp status - I agree that some more explanation of that will help here. I think I see why you write that, but it just needs to be fully clear to all readers and match more closely the conclusion at the end of the Abstract.

As to my own comments (please consider them of equal weight to the reviewer's - i.e., accept or reject & rebut as you think best):

- I think the point made in Discussion at lines 553+ (about long timelines to be expected for wetland mitigation projects) could be slipped into the abstract at line 45. Right now it is factual but not connected to other outcomes cited in the Discussion. Consider that many will not read past the Abstract.

- I find multiple spelling errors throughout the manuscript - please run a spell checker and then ensure that names are correct. For example: envrionment on line 194. Prevalance (line 197). communitiy (line 213). Paragraph following line 252: Diversitiy, Indix. These should be easy to catch with software or another reader.

- The Shannon-Weiner index is one step from providing more useful information - please see Jost, L. (2006). Entropy and diversity. Oikos, 113(2), 363-375 for more details, but H' needs to be expressed as e raised to H'. This should also alter the calculation of Pielou's J, which is after all a simple ratio of values. If you make this change your bird diversity data should make a bit more intuitive sense and be more current.

- thanks for providing the Supporting Information tables - my bet = people will use those in the future.

- an alternative interpretation for line 450 is that the shift toward woody species will slow down trajectories of change because they will spread more slowly than clonal graminoids etc. So then "reaching a dynamic equilibrium" may not be the case.

- line 463. Does a niche exist outside of an organism? I buy the argument that it does not, and so suggest that "many small niches and" could be omitted here.

- line 564. Nice last paragraph - I suggest ensuring that thoughts here convey well in the Abstract.

2. In your Methods section, please provide additional location information of the study site, including geographic coordinates for the data set if available.

3. We note that Figures 1, 2 and 3 in your submission contain [map/satellite] images which may be copyrighted. All PLOS content is published under the Creative Commons Attribution License (CC BY 4.0), which means that the manuscript, images, and Supporting Information files will be freely available online, and any third party is permitted to access, download, copy, distribute, and use these materials in any way, even commercially, with proper attribution. For these reasons, we cannot publish previously copyrighted maps or satellite images created using proprietary data, such as Google software (Google Maps, Street View, and Earth). For more information, see our copyright guidelines: http://journals.plos.org/plosone/s/licenses-and-copyright.

You may seek permission from the original copyright holder of Figures 1, 2 and 3 to publish the content specifically under the CC BY 4.0 license. 

If you are unable to obtain permission from the original copyright holder to publish these figures under the CC BY 4.0 license or if the copyright holder’s requirements are incompatible with the CC BY 4.0 license, please either i) remove the figure or ii) supply a replacement figure that complies with the CC BY 4.0 license. Please check copyright information on all replacement figures and update the figure caption with source information. If applicable, please specify in the figure caption text when a figure is similar but not identical to the original image and is therefore for illustrative purposes only.

Reviewers' comments:

Reviewer's Responses to Questions

**Comments to the Author**

1. Is the manuscript technically sound, and do the data support the conclusions?

Reviewer #1: Yes

2. Has the statistical analysis been performed appropriately and rigorously? 

Reviewer #1: N/A

3. Have the authors made all data underlying the findings in their manuscript fully available?

Reviewer #1: Yes

4. Is the manuscript presented in an intelligible fashion and written in standard English?

Reviewer #1: Yes

5. Review Comments to the Author

Reviewer #1: General comments

This is study of the long-term changes in plant communities of a 35-year old restored freshwater wetland, which is important because few studies have documented changes in freshwater vegetation for this long, particularly for woody plants. The study was carefully executed, the interpretation of the findings thoughtful, and the manuscript clearly written. Most comments I have are for clarification or to suggest improvements. No statistical tests are included, but they do not seem needed, because this is a descriptive observational study.

Specific comments

L80-81. This sentence seems out of place and could probably be deleted – the goal of the paper is not to guide how restoration should be assessed.

L93. Define turnover here and in the abstract.

L136-37. This sentence is not clear - was there peat in the restoration site already? Maybe this was supposed to be two sentences?

L138. Is the topographically restricted wetland the original wetland or the restored wetland?

L152-153. Explain how the meander survey worked. E.g., how many people, paths through the wetland, presence/absence or abundance? Also, was all vegetation sampling done the same way in 1992, 2002, and 2020?

L159. Is the 0.5 m2 quadrat 0.7x0.7 m square, 0.5x0.5 square (which would be 0.25 m2 in area), or a circular area of 0.5 m2 (if so what is radius as for the larger plots)?

L279 Table 1. Is Woody % correct or should it be number of woody species. If woody %, is it the percent of all species that were woody?

L281 and elsewhere. FAC species are considered to be hydrophytes by the USACE for use in identifying and delineating wetlands. E.g. see Lichvar, R.W. 2013. The National Wetland Plant List: 2013 wetland ratings. Phytoneuron 2013-49: 1–241. Published 17. July 2013. ISSN 2153 733. http://www.phytoneuron.net/2013Phytoneuron/49PhytoN-2013NWPL.pdf. Please justify the inclusion of FAC species as “Upland” species.

L322-328. List the abbreviations in the table and restate their meaning here to improve ease of reading.

Table 6, 7, 8, and other place where means are reported without SD or SE (also average CoC values in Tables 1, 2, 3). Please include +/- standard deviation or standard error, and number of observation (N) for any means. These statistics will strengthen the interpretation of results, and are especially important because no statistical tests are included. Either SE or SD is reported for birds, but please also include N and clarify it the measure of variation is SE or SD.

L402. It is not clear how the red maple swamp supports observations of beaver activity raising water level by 1 m. I would think an increase in 1m of water would reduce tree abundance.

L476-478. I don’t see how any of the findings presented in the paper affirm that species composition is approaching a dynamic equilibrium. This statement also seems to contradict others elsewhere about the dynamic wetland complex and shifts in the past 18 years (e.g. L 484-486).

L569-571. Explain the statement saying the wetland might not have achieved creation of scrub-shrub swamp - is this because there aren't enough shrubs?

6. PLOS authors have the option to publish the peer review history of their article (what does this mean?). If published, this will include your full peer review and any attached files.

Reviewer #1: **Yes: **Andrew Baldwin

---

## [Author Response · Author response to Decision Letter 0]

28 Apr 2021

All comments are addressed in the 'Response to Reviewers' document as requested in the Decision Letter.

---

## [Editor Report · Decision Letter 1]

3 May 2021

Successional dynamics of a 35-year old freshwater mitigation wetland in southeastern New Hampshire

PONE-D-21-04357R1

Dear Dr. McKown,

I am pleased to inform you that your manuscript has been judged scientifically suitable for publication and will be formally accepted for publication once it meets all outstanding technical requirements. Thank you for comprehensively handling the many other edits in your revision and detailing them well in your response.

I do have one lingering edit that can be fixed as you move forward - it is admittedly minor but will make one matter more clear and avoid embarassment for you later. I list it below under "Additional Editor Comments."

Kind regards,

David G. Jenkins, PhD

Academic Editor

PLOS ONE

Additional Editor Comments:

At lines 270-271: I think this is now confused, because Jost’s “effective diversity (D)” is actually e^(H’), so calling that term H’ is circular and does not work. I suggest changing this line to read “Diversity was calculated as effective diversity (D), which is based on the Shannon-Weiner diversity index (H’):”

This will also require that H’ be replaced by D at line 449.

Please also note how that sentence spells “diveristy” to correct the current misspelling.

---

## [Editor Report · Acceptance letter]

5 May 2021

PONE-D-21-04357R1 

Successional dynamics of a 35 year old freshwater mitigation wetland in southeastern New Hampshire 

Dear Dr. McKown:

I'm pleased to inform you that your manuscript has been deemed suitable for publication in PLOS ONE. Congratulations! Your manuscript is now with our production department. 

Kind regards, 

on behalf of

Dr. David G. Jenkins 

Academic Editor

PLOS ONE